# The Optical Response of a Mediterranean Shrubland to Climate Change: Hyperspectral Reflectance Measurements during Spring

**DOI:** 10.3390/plants11040505

**Published:** 2022-02-12

**Authors:** Jean-Philippe Mevy, Charlotte Biryol, Marine Boiteau-Barral, Franco Miglietta

**Affiliations:** 1IMBE-UMR CNRS 7263/IRD 237, Aix-Marseille Université, 13331 Marseille, France; charlotte.biryol@imbe.fr (C.B.); marine.boiteau-barral@etu.univ-amu.fr (M.B.-B.); 2Institute of Biometeorology, National Research Council (CNR-IBIMET), Via Caproni 8, 50145 Firenze, Italy; f.miglietta@gmail.com; 3IMèRA, Institut d’Etudes Avancées de l’Université Aix-Marseille, 2 Place Le Verrier, 13004 Marseille, France

**Keywords:** reflectance spectrum, drought adaptation, shrub, Mediterranean, plant–plant interaction, vegetation indices, climate change

## Abstract

Remote sensing techniques in terms of monitoring plants’ responses to environmental constraints have gained much attention during recent decades. Among these constraints, climate change appears to be one of the major challenges in the Mediterranean region. In this study, the main goal was to determine how field spectrometry could improve remote sensing study of a Mediterranean shrubland submitted to climate aridification. We provided the spectral signature of three common plants of the Mediterranean garrigue: *Cistus albidus*, *Quercus coccifera,* and *Rosmarinus officinalis*. The pattern of these spectra changed depending on the presence of a neighboring plant species and water availability. Indeed, the normalized water absorption reflectance (R975/R900) tended to decrease for each species in trispecific associations (11–26%). This clearly indicates that multispecific plant communities will better resist climate aridification compared to monospecific stands. While *Q. coccifera* seemed to be more sensible to competition for water resources, *C. albidus* exhibited a facilitation effect on *R. officinalis* in trispecific assemblage. Among the 17 vegetation indices tested, we found that the pigment pheophytinization index (NPQI) was a relevant parameter to characterize plant–plant coexistence. This work also showed that some vegetation indices known as indicators of water and pigment contents could also discriminate plant associations, namely RGR (Red Green Ratio), WI (Water Index), Red Edge Model, NDWI_1240_ (Normalized Difference Water Index), and PRI (Photochemical Reflectance Index). The latter was shown to be linearly and negatively correlated to the ratio of R975/R900, an indicator of water status.

## 1. Introduction

Many tools allow great advancements in the understanding of plant responses to their environmental constraints. Hyperspectral spectrometry, for instance, is an approach that makes it possible to understand the structure and functioning of ecosystems at different spatial scales (leaf, individual, canopy…) in a non-destructive condition [1]. Indeed, the spectral properties of plant cover both depend on the intrinsic characteristics of plants (morpho-anatomical structure, chemical composition, physiology, etc.) and the environmental conditions. Thus, each community or individual has a spectral signature which is the amount of energy emitted or reflected as a function of the wavelength [1]. Spectral signatures are directly linked to specific areas of the light spectrum. It is possible to focus on different pigments such as chlorophylls, carotenoids, and anthocyanins in the visible part of the spectrum (VIS, 400–700 nm), while in the near infrared (NIR: Near InfraRed, 700–1000 nm) or in short-wave infrared (SWIR: Short-Wave Infrared, 1000–2500 nm) regions, other information can be collected [2]. Although the NIR and SWIR regions contain little information on pigment composition [3], the SWIR region shows changes before visible symptoms appear. The NIR reflectance provides information on leaf tissue structure, LAI (Leaf Area Index), and leaf angular position at the canopy level, while SWIR reflectance is mediated by water and biochemical components such as cellulose, lignin and proteins. This implies that hyperspectral spectroscopy allows for the early detection of stresses suffered by organisms [2]. Thus, environmental stresses can cause physiological and structural changes, which in turn influence the spectral signature of the plant. It therefore appears that spectral analysis can detect these changes and can be used for a dynamic characterization of ecosystems functioning both in terms of biotic (invading species, herbivory…) and abiotic (water availability, post-fire regeneration…) alteration. Moreover, the novel advances in statistics, for instance, in multivariate analyses and machine learning, now allow us to deepen our knowledge on the optical response of plants to environmental stress. 

One of the fields of application for spectral analysis is remote sensing. The analyses are mainly centered on the use of spectral indices, which are calculated from a few selected wavelengths and correlated to specific morphological or physiological parameters, such as the water content. These indices are mainly focused on the visible and near infrared range, meaning a large part of the hyperspectral data are still unexplored. The Earth Explorer 8, FLuorescence EXplorer (FLEX) mission, the launch of which is planned for 2023 by the European space agency (ESA), aims to map vegetation on a global scale based on the fluorescence of chlorophyll [4]. This mission will provide valuable information on the dynamics of vegetation with a spatial resolution of 300 m [5]. The fluorescence that will be detected is the so-called solar-induced fluorescence (SIF), which is a passive measurement compared to the active fluorescence data derived from PAM (Pulse Amplitude Modulation) instruments [6]. Moreover, remote sensing data must be validated by ground measurements for a better understanding of the functioning of ecosystems considering plants’ biodiversity and the changes of environmental conditions.

Among the environmental stresses, drought appears to be one of the most worrying issues, at least as far as the Mediterranean region is concerned. Precipitation is concentrated in autumn and winter, while in summer, water resources are often in deficit. The latter is a determining factor for the growth, distribution, and diversity of plants [7,8,9]. In the context of climate change, these summer droughts could become more frequent, longer, and more intense [10]. In fact, there has been a trend of increasing temperatures and a decrease in precipitation during the last few decades in the Mediterranean region. Thus, it is possible to observe physiological alterations/variations of chemical properties of plants to overcome coexistence stress. There are, for example, species with trichomes, such as *Cistus albidus*, and others with thick cuticles and waxes, such as kermes oak. Several in situ precipitation manipulation experiments have been set up to study the consequences of these modifications on plant and ecosystem functioning [9,11,12]. It seems that the results of these experiments vary according to the composition and the richness of plant species [7,13]. A great plant richness seems to counterbalance the negative effects of aridification. In the context of various experiments, the response of ecosystems to climate change were identified, but few studies were carried out from natural shrub ecosystems [12,14,15]. To our knowledge, no experiment has been carried out on a scrubland dominated by *Quercus coccifera*, which is one of the most predominant shrubs in the Mediterranean region [16].

This in situ study was based on the CLIMED site (Impact of CLImate changes on biodiversity and its consequences on the functioning of a MEDiterranean ecosystem), an observatory installed in 2011 in a Mediterranean scrubland dominated by the kermes oak. The aim of this experimental site was to understand the variations in biodiversity and its consequences on the functioning of the ecosystem in response to the aridification of the climate. Site exclusion devices were designed to simulate a reduction of about 30% in precipitation, and the specific objectives of this work were: (i) to determine the spectral signature of three common species of Mediterranean shrubland (ii) to investigate how these signatures may vary depending on the diversity of plant community, (iii) and to determine how water availability may be optically characterized in terms of canopy structure and functioning.

## 2. Materials and Methods

### 2.1. The Study Area

The work was carried out on the CLIMED site, an observatory in shrubland (43°21′54″ N, 5°25′30″ E) located in the vicinity of the city of Marseille since 2011. The species which dominate this shrubland are white cistus (*Cistus albidus,* Ca), kermes oak (*Quercus coccifera,* Qc), rosemary (*Rosmarinus officinalis,* Ro), and gorse of Provence (*Ulex parviflorus,* Up). This site thus exhibited different natural combinations of plant associations according to the 4 species. In the frame of this work, only 3 species were considered since *Ulex parviflorus* was excluded due to a phenological mismatch: monospecific (Ro; Qc; Ca), bispecific (Ro-Ca; Ro-Qc; Ca-Qc), and trispecific stands (Ro-Ca-Qc). These different specific (seven) assemblages were selected in triplicate then spread under a set of devices, each covering a surface of 16 m^2^. Half of them had gutters that simulated the aridification of the climate by theoretically excluding 30% of rainfall. The other half were devices that acted as controls by having upside-down gutters (Figure 1). Therefore, the 7 combinations of plants selected represented a total of 42 plots. From 2014 to 2017, the average volumetric soil water content recorded (TDR100) under the exclusion devices at 10 cm deep in the soil was about 10–22%. 

### 2.2. Hyperspectral Signature Measurements

To mitigate the border effect, plants were selected at about 1 m from the edge of each plot. The measurements were carried out on 4 plants per plot on a clear day without strong wind between 10 and 15 h local time from May to June 2019.

The spectral reflectance of the canopy was measured using a portable field spectroradiometer (ASD FieldSpec No. 4, USA), which can detect the spectral signature of the canopy in the range of 350–2500 nm with a width sampling rate of 1.4 nm. Three detectors enabled the spectrum to be recorded: visible and near IR (350–1000 nm) with a spectral resolution (RS) of 3 nm using a silica iodine array containing 512 photodiodes. Short wavelengths in the infrared SWIR 1 (1001–1800 nm) with RS of 10 nm and short wavelengths in the infrared SWIR 2 (1801–2500 nm) with RS of 10 nm were recorded. The ASD spectrometer sensor was operated with a field of view of 25° at 10 cm above the plant which corresponded to a diameter of 4 cm for this field in order to minimize the influence of soil.

The instrument calculates the reflectance as the ratio of the incident radiation reflected by the plant versus the incident radiation reflected by a white reference Spectralon. All the measurements were carried out with the sensor pointed at nadir on the target, and 3 successive measurements were carried out on each target to check for sensor stability. The white panel calibration was carried out before and after each plant reflectance acquisition.

View Spec Pro (ASD) software was used to preprocess and extract reflectance spectra. Certain wavelength ranges strongly disturbed by noise due to the intense absorption of atmospheric water were removed from the data to be processed: from 1340 to 1430 nm, 1810 to 1980 nm and beyond 2400 nm.

The spectral signatures of all species were then analyzed under association (monospecific, bispecific and trispecific) and drought conditions (control and rain exclusion).

### 2.3. Plant Water Status from Spectral Reflectance: The Ratio of R975/R900

Plants’ reflectance spectra exhibited several water absorption bands among the band from 920 to 1110 nm [17]. In this study, the trough was identified at 975 nm and normalized at 900 nm where water was absorbed less strongly, and also because the sample presentation was affected in the same way as suggested in [18].

### 2.4. Vegetation Structure and Stress Indices

Various indices related to vegetation and reflectance were applied to the data set (Table 1). The PRI (Photochemical Reflectance Index) detects changes in the pigments of the xanthophyll cycle (pigment belonging to the carotenoid family), and it is also considered as a proxy of Non-Photochemical Quenching (NPQ). The WI (Water Index) was used to estimate the water content of individual plants and thus, to highlight the effect of drought on all the plant communities studied.

### 2.5. Scanning Electron Microscopy (SEM)

Leaves of Ca, Qc, and Ro were collected for micromorphological observations of their surface structure. For each species, 5 leaves were examined on both adaxial and abaxial faces. These were gold coated prior to scanning through an electronic microscope: FEI XL30 ESEM (USA).

### 2.6. Statistical Analysis

A total of 780 reflectance measurements were collected, imported to Metaboanalyst [31], where the spectra were normalized centered and reduced, and then processed by multivariate analysis (Discriminant Analysis of Partial Least Squares: PLS-DA) and ANOVA followed by the LSD post hoc test. The aim was to discriminate the plant species (Ca, Qc and Ro) based on their spectral signature and to identify a potential effect of aridification in plant communities (monospecific vs. bispecific vs. trispecific). The most discriminating variables were subsequently identified according to their VIP score (Variable importance in projection). The data were processed with StatGraphics for the interaction and correlation tests.

## 3. Results

### 3.1. Species Spectral Discrimination

Discriminant Analysis of Partial Least Squares (PLS-DA) clearly distinguished Ca from the other two species (Figure 2). The total inertia of the factorial plane of axes 1 and 2 was 93.9%, explaining almost all the observed variability. The mean of reflectance spectra of each species was calculated to determine their spectral signature. Ca exhibited a profile that differed from Qc and Ro over all the spectrum (350–2400 nm), and also had a higher reflectance (Figure 3). The analysis of variance (ANOVA) carried out showed that the spectral signatures were significantly different (ANOVA, *p* < 0.001); the subsequent LSD test allowed us to compare the species in pairs at each wavelength and to identify the most discriminant spectral bands in the three spectral zones, namely the visible (VIS), the near infrared (NIR), and the short wavelengths of the infrared (SWIR) (Table 2). The significant wavelengths in each band (VIS, IR, and SWIR) could explain the physiological and structural origin of the main differences existing between these species. From Ca, in the visible, this corresponds to the bands from 448 to 534 nm, which is characteristic of blue or blue-green and 563–700 nm. In the near infrared, the wavelengths obtained vary from 701–100 nm, which may be related to the cellular structure. In the short wavelengths of the infrared, several bands were characterized among these: 2204–2400 nm, which include the water absorption band. It is therefore possible to discriminate Ca from the two other species in all areas of the spectrum whether in the visible, near infrared, or mid infrared region.

Ca, Qc, and Ro only showed differences in their spectrum in certain bands of the visible and short wavelengths (SWIR): from 350 to 396 nm, from 414 to 447 nm (chlorophyll a, b, and β-carotene absorption bands), and from 536 to 561 nm. The latter range corresponded to the reflectance of wavelengths which were weakly absorbed by plants. The ranges of 2160–2203 nm and 2147–2148 nm also include the water absorption band.

### 3.2. Scanning Electron Microscopy (SEM)

Variations in the morphoanatomical structure on the leaf surfaces of the plant studied were found using a scanning electron microscope (Figure 4). Once again, the leaf surface of the Ca was particularly dense in ornamentation. A large network of trichomes was observed on the entire leaf surface (Figure 4A,D). From leaves of Ro, the adaxial side was covered by crystal waxes and scattered glandular trichomes (Figure 4B), while tector and glandular trichomes were located mainly on the midrib of the abaxial side (Figure 4E). Finally, only a few scattered hairs were identified on the adaxial faces of Qc (Figure 4C), but it is interesting to notice the high density of stomata poorly surrounded by tector trichomes on the abaxial side (Figure 4F).

### 3.3. Hyperspectral Response of Interspecific Assemblages in Rain Exclusion Conditions

While the three species were well separated according to their spectra, it was relevant to see if the spectral behavior of a species could change according to the presence of another species. The comparison of the various spectra of species in different assemblages shows that the reflectance varied according to the assemblage considered (Figure 5). In control conditions, a decrease in the reflectance all over the spectrum was observed when Ca coexisted with Ro (cCaRo) (Figure 5B) compared to the monospecific stand of Ca (cCa) (Figure 5A). For instance, at 1100 nm, the reflectance dropped from 0.45 to 0.37. For Qc, a significant increase in the reflectance was observed in trispecific assemblage (cQcCaRo) (Figure 5D,F). When comparing control versus exclusion assemblages, it appears that there was no effect of rain exclusion in monospecific stands (Figure 5A,D,G), while in trispecific assemblages (eCaQcRo; eQcCaRo; eRoCaQc), the reflectance seemed to decrease (Figure 5C,F,I).

### 3.4. The Ratio of R975/R900, Plant Response to Drought and the Effect of Interspecific Assemblages

The objective of this part of the study was dual: on the one hand, we aimed to find out whether aridity could be revealed by optical measurements and, on the other hand, we aimed to determine whether the effects of the same aridity were modified by interspecific interactions. From the mean spectra of each species, important differences occurred in the near IR (Figure 5). The comparisons were therefore realized considering the water absorption band at 975 nm after normalization (R975/R900 nm).

From the two-way-ANOVA, a significant effect of plants’ assemblage (*F* = 52.11; *p* < 0.001) and rain exclusion (*F* = 18.88; *p* < 0.001) was shown. However, a significant interaction existed between these two factors (*F* = 4.57; *p* < 0.001). Thus, the normalized reflectance R975/R900 was first tested by water treatment for all 12 of the plant combinations (Figure 6). Concerning the control devise, the main objective was to check whether plant water content varied depending to the species mixing (Figure 6A). As a result, no effect on water content was observed from Ca except in the presence of Ro (CaRo). Conversely, the reflectance of Ro in the presence of Ca (RoCa) increased significantly compared to the monospecific stand, which resulted in a lower water content from Ro in such bispecific assemblages. A significant increase in the reflectance from Qc in the trispecific assemblage (QcCaRo) was obtained compared to the monospecific stand. In the rain exclusion condition (Figure 6B), a significant drop in the reflectance of Qc associated with Ca and Ro (eQcCaRo) was noticed, as well as that of Ro in the trispecific assemblage (eRoCaQc). For a second time, the normalized reflectance R975/R900 was tested by plant assemblage for the control and water exclusion device conditions (Figure 7). It appears that for all three of the species in trispecific associations, the reflectance in rain exclusion condition decreased significantly (11–26%). For bispecific assemblages, only the reflectance of Ro associated with Ca (RoCa) decreased when part of the precipitation was excluded compared to the controls.

### 3.5. Vegetation Indices, Drought, and Interspecific Assemblages

Several vegetation indices mostly related to plant water content were also tested. The PLS-DA analysis (Appendix A) showed once again that Ca differed significantly from Qc and Ro, regardless of the plant association in which it was found and the drought condition. The total inertia was about 78% on axes 2 and 1, with the latter representing a major part of the variance (57%). Two groups were clearly displayed; the group of Ca was negatively correlated to axis 1, while the other group consisting of the biotic interactions of Qc and Ro was positively correlated to the same axis. The VIP (Variables In Projection on axis 1) analysis performed showed that nine vegetation indices’ VIP scores were considered as important for the model; among these were NPQI, WI/NDVI, SIFB, and NDI (Appendix A). The pheophytinization index (NPQI) appeared to be the most discriminate variable for species assemblage. Hence, two-way-ANOVA was carried out to examine the dual effect of the assemblage and drought based on vegetation indices. While all the indices enabled the characterization of plant–plant coexistence, only RGR, WI, PRI, Red Edge Model, and NDWI_1240_ exhibited significant effects in both drought and plant assemblage (Table 3). Except for WI, only PRI was significantly correlated with the R975/R900 ratio (Figure 8).

## 4. Discussion

Today, plant biodiversity constitutes major challenges in the context of global change. Ground-based measurements of ecosystems’ biodiversity and functioning monitoring are crucial for further spatial data acquisition from remote sensing either by air-born or satellite. The PSL-DA analysis carried out clearly allowed us to discriminate the spectral footprint of *C. albidus* from those of *Q. coccifera* and *R. officinalis* throughout the spectrum from 350 to 2500 nm (Figure 2). The use of field spectrometers was shown to be a powerful tool in discriminating cultivars [32,33] as well as plant varieties [34] and closely related species [35]. The differences observed are related to both canopy structure, leaf optical, and biochemical properties. In the visible domain, the leaf reflectance spectrum is determined by photosynthetic pigments [36]. The shape of the *C. albidus* spectrum in this region is characterized by a typical shoulder from 400 to 500 nm. In the latter spectral band, pigments such as chlorophyll a (430 nm), chlorophyll b (445 nm) and carotenoids (450 nm) exhibited maximum absorption values. As a result, the high reflectance of *C. albidus* in this region may be interpreted as a weak absorption of chlorophylls and carotenoids in accordance with the blue-green appearance of the leaves. Similarly, the three species spectra exhibited differences between 536 and 561 nm that corresponded to a lesser absorption rate of green light by chlorophylls and carotenoids (Table 2; Figure 3). Indeed, increasing chlorophyll *a* concentration is inversely proportional to the reflectance of green light [37]. Therefore, it is likely that green light became diffusely reflected from as structures such as cell wall components. In the NIR region (700–1500 nm), reflectance changes are generally associated with light scattering in the leaves and between leaves of the canopy and also with water content [36,38,39]. Typically, this concerns leaf tissue structure, area index, and angle distribution within the canopy. The sharp increase in reflectance from 700 nm, also described as the red edge, is the consequence of the loss of pigment absorption. Two bands of water absorption are shown around 970 and 1200 nm. In the shortwave-infrared (1500–2500 nm) region, the three species could be distinguished at about 2200 nm, which highlighted C–H bonds’ vibrational spectra in proteins, sugars, and cellulose [40]. In addition to intracellular leaf components, external reflective features may modify the biophysical properties of leaves. The high density of trichomes found in *C. albidus* leaves’ SEM pictures (Figure 4A,D) may also explained the differences of reflectance obtained compared to the two other species (Figure 5A,D,G). Indeed, from the leaf characteristics of four plant species in terms of color, microscopic surface structures, and thickness, the highest reflectance levels were recorded throughout the spectrum from the hairy leaves of *Cineraria maritima* [41].

This work also showed that interspecific assemblage may be characterized optically, mainly in the NIR band (Figure 5). In trispecific stands, *Rosmarinus officinalis* showed a lower reduction in reflectance than other species in the rain exclusion condition. This may be explained by the smallest leaf exchange surface with the environment (Figure 5C,F,I).

We noticed an overall trend of the decrease in reflectance in bispecific assemblages compared to monospecific stands. This may be interpreted as an increase in the optical absorption of metabolites, water, as well as changing LAI-dependent optical scattering and leaf angle. Early potted experiments revealed that secondary metabolites’ synthesis and/or accumulation may vary according to plant assemblage. Indeed, the emission of terpenes from *C. albidus* was reduced when assembled either with *R. officinalis* or *Q. coccifera*, whereas no changes were observed from *R. officinalis* and *Q. coccifera* in the different bispecific assemblages [42]. However, when *Pinus halepensis* was in competition with *R. officinalis*, this resulted in an increase in the accumulation of monoterpenes. In contrast, during trispecific plant–plant interference, the canopy reflectance tended to increase (Figure 5), probably as the result of intense competition for the resources. Hence, it is likely that natural interspecific competition/facilitation could lead to the biosynthesis and storage of target molecules. In this regard, the ability to remotely characterize the occurrence of specific phytochemicals as the response of a plant submitted to biotic or abiotic stress has also been demonstrated [43,44]. The creation of beneficial or detrimental effects to other neighboring plants when plants coexist is a common scenario, known as the allelopathy effect. These allelochemicals can increase spectral absorbance.

We thus tried to understand to what extent the spectral analysis could highlight the impact of the aridification of the climate on interspecific assemblages based on the reflectance ratio of R975/R900. In gerbera plants, the ratio R970/R900 was shown to be linearly correlated with leaf water status [18]. Numerous experiments carried out on plants subjected to water stress have shown that reflectance increases with the intensity of drought [45,46,47]. On one hand, at the level of the control devices, we observed a significant drop of reflectance in the water absorption band only for CaRo compared to its monospecific stands (Figure 6A). On the other hand, the reflectance from *R. officinalis* in the presence of *C. albidus* (RoCa) increased. These results may be interpreted as a competition for water resources, which is detrimental to rosemary. In the same way, the increase in the R975/R900 ratio from QcCaRo suggests that *Q. coccifera* water demand is not satisfied in such a trispecific association, probably because of its high stomata density and lower limit layer resistance (Figure 4F). The question that arises that must be understood is how this competition will evolve with the reduction in precipitation expected in the Mediterranean basin. Surprisingly, we observed a decrease in the reflectance in all trispecific combinations with rainfall exclusion, suggesting that more diversified species communities will better resist the predicted climate aridification (Figure 7). For CaQcRo, the improvement of cistus water content may be interpreted as the result of synergic interaction between *Q. coccifera* and *R. officinalis* since no effect was recorded in bispecific assemblies (CaRo; CaQc). Similarly, the interaction between *C. albidus* and *R. officinalis* may explain the increase in water content from the leaves of *Q. coccifera* in trispecific combinations (QcCaRo). However, for *R. officinalis* in a trispecific association (RoCaQc), the presence of *C. albidus* seems to play a crucial role in water saving, since the ratio of R975/R900 was significantly low for RoCa. These results underline the complexity of biotic interactions, including those between plants. It is likely that these interactions involve chemical mediators at both the root and leaf level. *C. albidus* and *R. officinalis* are both emitters of volatile organic compounds which can affect the physiology of surrounding species. Indeed, it has been shown that oak, which is a strong emitter of isoprene, could improve the resistance of maple in the context of moderate drought [48]. More generally, we observed a decrease in reflectance in all trispecific combinations with rainfall exclusion, suggesting that more diversified species communities will better resist the predicted climate aridification. This will probably be achieved by developing a more sophisticated water-saving mechanism.

The multivariate analysis carried out from 17 vegetation indices allowed us to identify 9 explicative variables (VIP scores) of the species distribution, principally: NPQI and WI/NDVI. However, NPQI, which corresponds to pigment degradation through the occurrence of phaeopigments, was shown to increase with drought severity [49]. This disagreement with our results could be explained by the moderate severity of the drought conditions applied under our exclusion systems. Most vegetation indices have no direct link with photosynthetic machinery. We found that the proxies of solar-induced fluorescence in the oxygen A and B absorption bands are discriminant variables, suggesting that plant association could be detected remotely based on that chlorophyll emission signal. Additionally, the photochemical reflectance index (PRI), which gives an estimation of PSII efficiency and net CO_2_ uptake, was linearly correlated to the ratio of R975/R900. This is in accordance with the previous work arguing that PRI was a relevant indicator of drought stress in several European shrublands [22].

In conclusion, this study demonstrated the strength of hyperspectral measurements both in terms of species discrimination and their responses to environmental constraints such as drought. We characterized the spectral signature of three species mainly found in Mediterranean shrublands: *C. coccifera, C. albidus*, and *R. officinalis*. These spectral profiles changed depending on the neighboring species and drought conditions. These data will allow for the implementation of remote sensing tools for dynamic monitoring of shrubland as a function of environmental constraints such as drought and post-fire regeneration. However, the temporal variability of the hyperspectral response and the physiological interpretation in terms of remote sensing of those changes still remains challenging. Indeed, little is known about the role and production of key chemicals of plants in response to competing neighbors. Further phenological, ecophysiological, and chemometric studies are required to fill this knowledge gap of the optical response of plants to environmental constraints.

## Figures and Tables

**Figure 1 plants-11-00505-f001:**
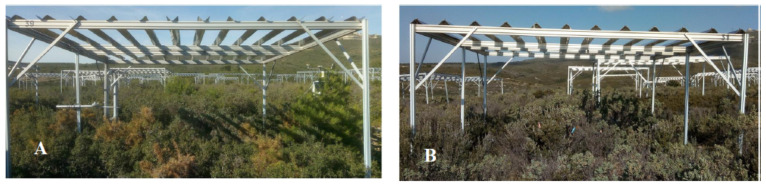
The site CLIMED: rain exclusion (**A**) and control devices (**B**).

**Figure 2 plants-11-00505-f002:**
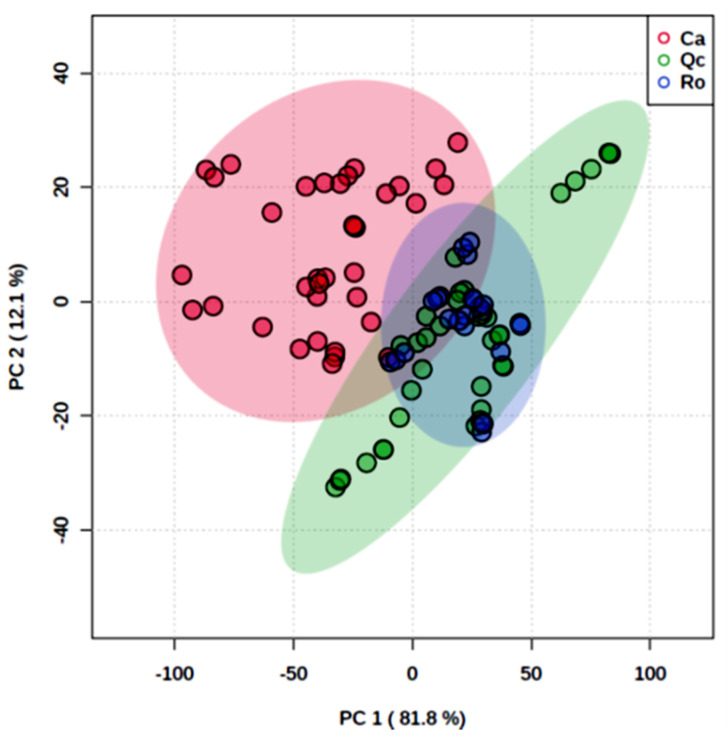
Discriminant analysis of partial least squares (PLS-DA) of *Cistus albidus* (Ca) *Rosmarinus officinalis* (Ro) and *Quercus coccifera* (Qc) in monospecific populations. Plan of individuals on axes 1 and 2.

**Figure 3 plants-11-00505-f003:**
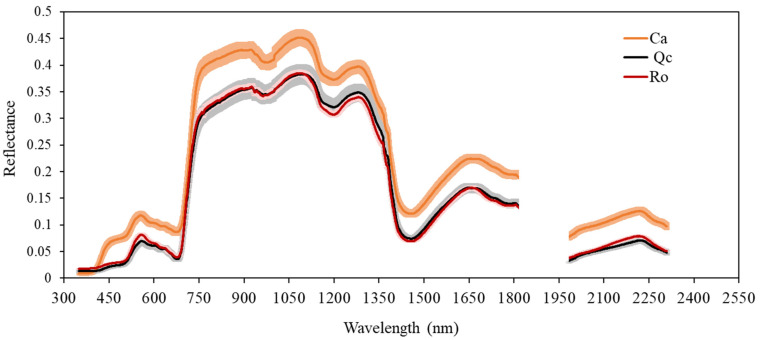
Spectral signatures of *Cistus albidus* (Ca), *Quercus coccifera* (Qc) and *Rosmarinus officinalis* (Ro)in monospecific populations. Means with standard errors. *n* = 12. The hole corresponds to water absorption band.

**Figure 4 plants-11-00505-f004:**
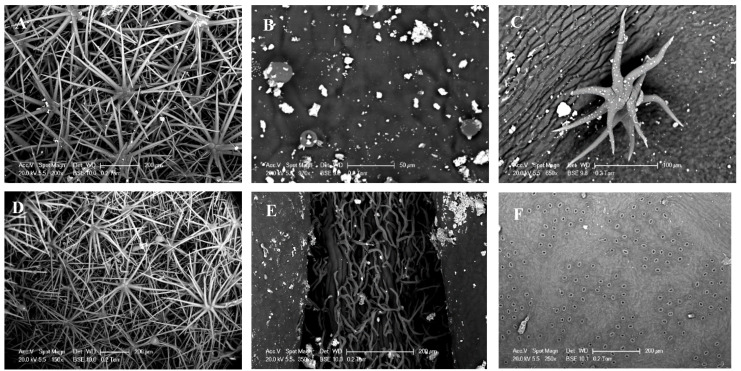
Scanning electron microscopy (SEM) of the leaf adaxial side of *Cistus albidus* (**A**), *Rosmarinus officinalis* (**B**) and *Quercus coccifera* (**C**). The abaxial sides: (**D**–**F**), respectively.

**Figure 5 plants-11-00505-f005:**
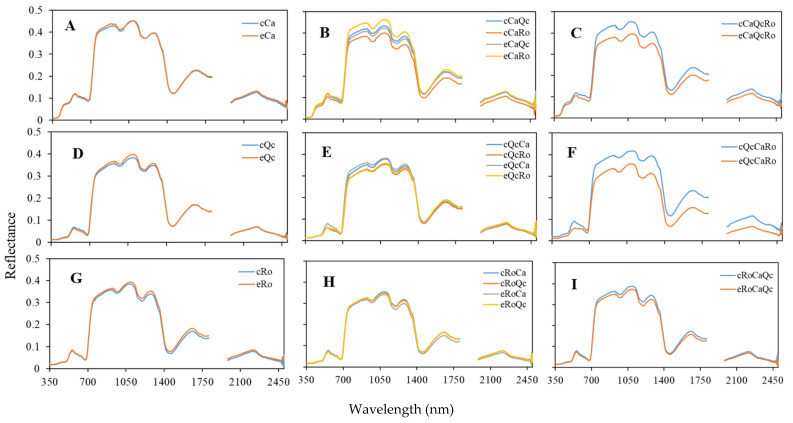
Spectral reflectance and specific assemblages in control (c) and rain exclusion (e) conditions of *Cistus albidus* (Ca) (**A**–**C**); *Quercus coccifera* (Qc) (**D**–**F**); and *Rosmarinus officinalis* (Ro) (**G**–**I**) in monospecific (Ca; Qc; Ro), bispecific (CaQc; CaRo; QcCa; QcRo; RoQc) and trispecific (CaQcRo; QcCaRo; RoCaQc) stands. *n* = 12. The first letter in majuscule indicates the genus of the species from which the measurement was carried out.

**Figure 6 plants-11-00505-f006:**
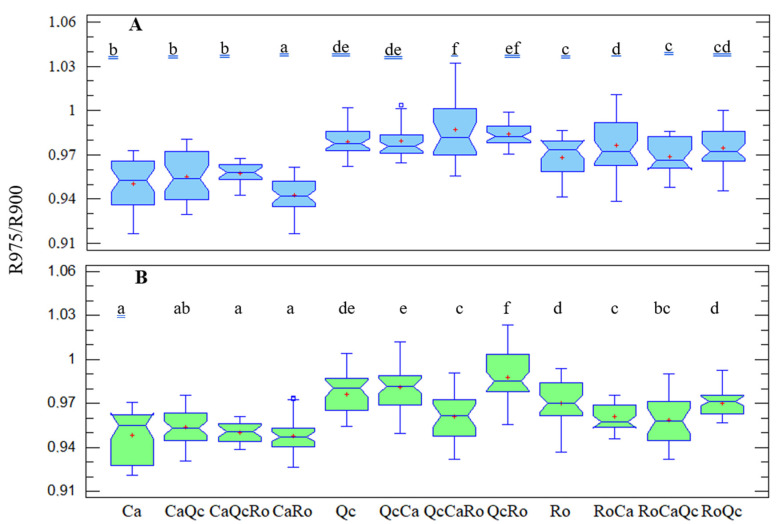
Notched boxplot of the ratio of R975/R900 of *Cistus albidus* (Ca) *Rosmarinus officinalis* (Ro) and *Quercus coccifera* (Qc) in monospecific (Ca; Qc; Ro), bispecific (CaQc; CaRo; QcCa; QcRo; RoQc), and trispecific (CaQcRo; QcCaRo; RoCaQc) stands. (**A**) Under control devices and (**B**); under exclusion devices. ANOVA, *p* < 0.05 followed by LSD tests. Significant differences are shown by different letters. a < b < c < d < e < f. *n* = 12. The first letter in majuscule indicates the genus of the species from which the measurement was carried out.

**Figure 7 plants-11-00505-f007:**
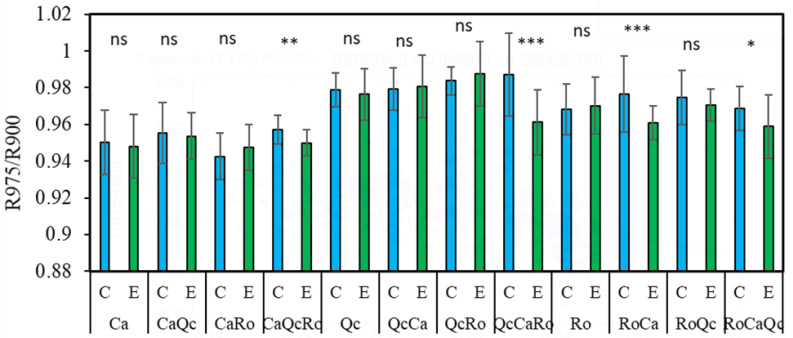
The ratio of R975/R900 of *Cistus albidus* (Ca) *Rosmarinus officinalis* (Ro), and *Quercus coccifera* (Qc) in monospecific (Ca; Qc; Ro), bi-specific (CaQc; CaRo; QcCa; QcRo; RoQc), and trispecific (CaQcRo; QcCaRo; RoCaQc) stands. C; control devices and E; exclusion devices. ANOVA, *p* < 0.005 *; *p* < 0.001 **; *p* < 0.001 ***. ns; non significative. *N* = 12. The first letter in majuscule indicates the genus of the species from which the measurement was carried out.

**Figure 8 plants-11-00505-f008:**
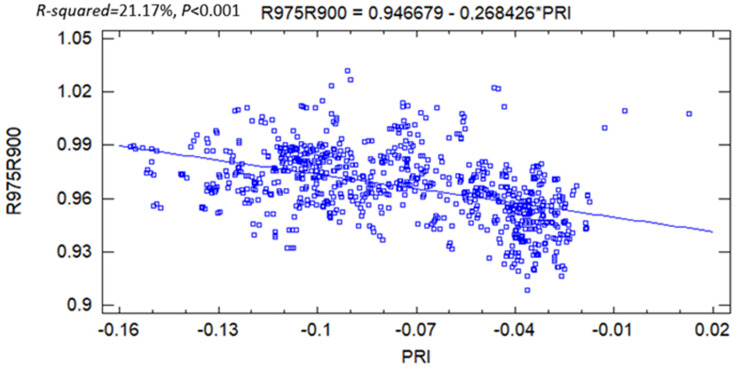
Correlation curve between the ratio of R975/R900 and PRI.

**Table 1 plants-11-00505-t001:** Common vegetation indices found in the literature.

Index	Abbreviation	Formula	References
Photochemical Reflectance Index	PRI	(R531−R570)/(R531 + R570)	[19]
Normalized Pheophytinization Index	NPQI	(R415 − R435)/(R415 + R435)	[20]
Water Index	WI	R900/R970	[21]
Normalized Difference Vegetation Index	NDVI_680_	(R780 − R680)/(R780 + R680)	[22]
	NDVI_570_	(R780 − R570)/(R780 + R570)	
	NDVI_chlorophyll_	(R800 − R680)/(R800 + R680)	
Normalized Difference Water Index	NDWI_1240_	(R860 − R1240)/(R860 + R1240)	[23,24]
	NDWI_2130_	(R860 − R2130)/(R860 + R2130)	
Pigment Sensitive Normalized Difference 2	PSND2	(R800 − R635)/(R800 + R635)	[25]
Difference Index	DI1	R800 − R550	[26]
Red Green Ratio	RGR	(R612 + R660)/(R510 + R560)	[27]
Normalized Difference Index	NDI	(R700 − R670)/(R700 + R670)	[28]
Solar Induce Fluorescence proxies	SIF	SIFB (R685/R850) and SIFA (R740/R630)	[29]
Red Edge Model	REM	[(R800/R700) − 1]	[30]

**Table 2 plants-11-00505-t002:** Wavelengths (nm) specific to each plant and common to the 3 species identified by one-way ANOVA (*p* < 0.001) and post hoc analysis (Fischer’s LSD test) in VIS, NIR, and SWIR from pairwise comparison of *Cistus albidus* (Ca), *Quercus coccifera* (Qc) and *Rosmarinus officinalis* (Ro) spectra. *n* = 12.

Species	Visible (VIS)	Near Infrared (NIR)	Short Wavelenght of the Infrared (SWIR)
Ca compared to Qc and Ro	448–534 563–700	701–1000	1001–1376 1408–2159 2175 2204–2400
Qc compared to Ca and Ro	408–413		
Ro compared to Ca and Qc	397–406		
Common to Ca, Qc and Ro	350–396 407 414–447 535–562		1377–1407 2147–2148 2160–2174 2176–2203

**Table 3 plants-11-00505-t003:** Two-way ANOVA of the different vegetation indices investigated. Factors of water treatment and species assemblage. Degrees of freedom between water treatment (control vs. rain exclusion) range to 1 and within groups of assemblages (monospecific of Ca, Qc, and Ro, bispecific of CaQc, CaRo, QcCa, QcRo, RoCa, and RoQC, and trispecific of CaQcRo, QcCaRo, and RoCaQc) at 11.

Vegetation Indices	Water Treatment		Assemblage	
	F-Values	*p*-Values	F-Values	*p*-Values
PRI	4.06	0.044	159.61	<0.001
NPQI	0.09	ns	1365.02	<0.001
WI	10.83	<0.01	41.1	<0.001
NDVI_680_	0.84	ns	162.37	<0.001
NDVI_570_	0.05	ns	60	<0.001
NDVI_chlorophyll_	1.01	ns	170.45	<0.001
NDWI_1240_	11.1	<0.001	39.13	<0.001
NDWI_2130_	0.8	ns	40.87	<0.001
PSND2	1.79	ns	8.49	<0.001
DI1	1.12	ns	13.69	<0.001
RGR	14.11	<0.001	27.92	<0.001
NDI		ns	307.72	<0.001
SIF(A)	2.16	ns	75.55	<0.001
SIF(B)	1.35	ns	163.14	<0.001
REM	3.93	0.047	26.63	<0.001
WI/NDVI	2.44	ns	222.29	<0.001

## Data Availability

Not applicable.

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
