# Peer review of "The Optical Response of a Mediterranean Shrubland to Climate Change: Hyperspectral Reflectance Measurements during Spring"

_plants, 2022, doi:10.3390/plants11040505_

Round 1

Reviewer 1 Report

Studies on the spectral properties of plants to monitor their responses owing to environmental constraints become popular and, are essential as we do not need to carry out chemical analysis to identify the plant stresses. Most importantly, we can detect stresses of plants even before appearing the visible symptoms. However, the following comments should be considered before publishing the paper.

Introduction

  • Page 2, line 45 – We can obtain spectral data related to pigments of plants using the visible part of the spectrum. Briefly explain about the information that can be obtained at infrared and short-wave infrared ranges.
  • Page 2, lines 77–79 – You have described the morphological difference of plant leaves in response to drought stress. It is better to mention physiological alterations/variations of chemical properties of plants to overcome coexistence stress.
  • Page 2, line 66 – Define “PAM”.
  • Page 2, line 89 – Define “CLIMED”.

Please define the abbreviations on the first use even though if it is a common term.

 Materials and methods

  • Page 3, line 107 – Why did “Ulex parviflorus” exclude? Is that a less common plant in this region than other plants?
  • Page 4, line 144 – “Vegetation structure and stress indices” Is this a sub topic?

Results and Discussion

  • Figure 5 – In tri-specific stands, Rosmarinus officinalis showed a less reduction of reflectance than other species at rain exclusion condition. Briefly explain the possible reasons in discussion section.
  • Page 9, lines 309–310 – In species spectral discrimination study, a clear deviation of reflectance were observed in Cistus albidus compared with other two species due to high density of trichomes on leaves. You have compared spectral properties of plants based on aridification and plant coexistence stress. Briefly explain how could you identify/differentiate the reflectance data given from external reflective features or intracellular components of leaves? Figure 5 will be helpful to discuss it.
  • The creation of beneficial or detrimental effects to other neighboring plants when plants coexist is a common scenario, known as allelopathy effect. These allelochemicals can increase spectral absorbance.
  • Try to follow a standard method for reporting F statistics (ANOVA results). Consider degrees of freedom of between and within groups when reporting results.

Abstract

  • In abstract, uncommon abbreviations should be avoided unless they are essential. Since abstract should be able to stand alone, abbreviations must be defined at their first mention.

 FORMATTING/TYPOS ERRORS

  • Page 1, line 20 – “11-26%”, it should be an “en dash” “11–26%”.  En dash is normally used to mark ranges. Modify here and throughout.
  • Page 4, line 162 – “potentiel” should be “potential”.
  • Page 6, table 2 – “wavelenght” should be “wavelength”.
  • Page 10, line 296 – “bleu-green” should be “blue-green”.
  • Page 12 line 367 – “lettres” should be “letters”.

Reviewer 2 Report

This manuscript by Mevy et al. used hyperspectral reflectance measurements to study the optical response of a mediterranean shrubland to climate change. The multivariate analysis carried out from 17 vegetation indices allows to identify 9 explicative variables (VIP scores) of the species distribution, and such kind of systematic comparison is really meaningful and interesting. The manuscript is well organized and clearly presented, it could be accepted after a few minor changes. The following list my detailed comments on the manuscript.

  1. Lines 141: the sentence is problematic: “these the ”. Also, the physical reasons for the treatments are suggested to be briefly mentioned.

  1. From Table 1, it seems that only reflectance at a couple particular channels are used, would hyperspectral observations still necessary?

  1. the dots in Figure 8 are almost randomly distributed. Although a linear fit is given, how is the correlation coefficients between the two parameters, would such fitting reasonable?

  1. It is noticed that only observations during spring were discussed. is it possible or necessary to extend this study for other seasons?

Reviewer 3 Report

   The work by Mevy et al. is devoted to revealing relations between plant drought and plant reflectance in common plants of the Mediterranean garrigue. The work seems to be interesting; however, there are minor comments and remarks.

  1. Introduction: Authors analyzed not only water reflectance indices; vegetation and pigments indices were also analyzed. I suppose that earlier works, which analyzed the influence of the drought on these vegetation and pigments indices, should be noted in Introduction. For example, drought can induce fast changes in the photochemical reflectance index (PRI).
  2. Section 2.1: Cultivation of plants should be described in more details. E.g., were vegetation pots used or were plant cultivated in field conditions? Were natural precipitations? Etc.
  3. Order of figures should be checked (e.g., Figure 8 was placed before Figures 6 and 7).
